# An improvement of Q-Learning based on attenuation oscillation curve for path planning

Shiwen Sheng
Navigation College
Dalian Maritime University
Dalian, China
bak2220193276@dlmu.edu.cn

Yi Zuo
Navigation College
Dalian Maritime University
Dalian, China
zuo@dlmu.edu.cn

Yuzhou Lu
Navigation College
Dalian Maritime University
Dalian, China
yuzhoulu1201@163.com

Wei Wu
College of Science
Liaoning University of
Technology
Jinzhou, China

*Abstract—Reinforcement Learning can be applied to many fields, but it is still a problem how to balance exploration and exploitation it reasonably in the strategy of action selection. In order to solve the problem of path optimization in unknown environment, a model based on Q-Learning algorithm is proposed to balance the exploration stage and the exploitation stage. In this paper, based on Q-Learning algorithm, attenuation oscillation curve combined with it. The algorithm can reasonably adjust the time allocation of exploration stage and exploitation stage according to the current iteration times and iteration states. Effectively avoid the problem of over- exploitation and over-exploration of algorithms. In the path optimization simulation experiment, Q-Learning algorithm based on oscillation attenuation is compared with classical Q-Learning algorithm. Experiments show that for the same environment full of obstacles, Q-Learning algorithm based on oscillation attenuation can carry out path planning with better number of steps in a shorter search time, and has the ability to jump out of the local optimal.*

*Keywords—Exploration/Exploitation, Attenuation oscillation curve, Q-Learning algorithm, path planning*

## I. INTRODUCTION

Reinforcement learning has received extensive attention, such as game theory, operations research, cybernetics, simulation optimization and so on. But some challenges come along. One of the challenges that arises in reinforcement learning is not other types of learning, but the tradeoff between exploration and exploitation. In order to get a large number of rewards, the agent must choose the behavior that it has tried in the past and found to be effective in generating rewards. But to discover such a behavior, it must try a behavior it hasn't chosen before. In order to be rewarded, the agent must take advantage of what it has already experienced, but it must also explore in order to make better choices of action in the future[1]

Among them, reinforcement learning is applied to the optimization of path planning. The purpose of motion planning is to find a feasible path from the initial state to the target region in the configuration space without colliding with any obstacles[2]. At present, the main algorithms applied to path planning include artificial potential field method[3], viewable method[4], particle swarm optimization algorithm[5], AMP-PSO[6] and so on. In this paper, Q-Learning[7] algorithm of reinforcement learning is applied to path planning. Q-Learning algorithm is a classical reinforcement learning algorithm. It uses

Markov decision process (MDP) to describe the path planning problem[8]. For example, Q-Learning algorithm is applied to fuzzy output feedback tracking control problem[9].

Many researchers have improved the balance between exploration stage and exploitation stage in Q-Learning algorithm. Maozu Guo applied the criteria in the simulated annealing Algorithm to the SA-Q-Learning algorithm for exploration and exploitation[10]. It allows the agent to choose an action other than the strategy currently learned, and the probability of choosing the action increases with the increase of the proximity between its Q value and the optimal action Q value, that is, the probability of the agent choosing an action closer to the optimal action is greater than that of other actions. In addition, with the agent's continuous interactive learning of the environment, the set temperature parameter T approaches 0, and the algorithm approaches convergence. In addition, a number of other approaches have been applied to the exploration and exploitation balance of reinforcement learning, such as many approaches that utilize counters[11], model learning [12], or reward comparisons in a biologically inspired way [13].

The ε-greedy[14] is a common algorithm in Q-Learning algorithm. When selecting an action, the agent has a certain probability to explore the new environment, instead of always following the learned strategy. Compared with the traditional Q-Learning algorithm, ε-greedy has a certain probability to find the optimal action, but with the increase of learning time, the algorithm still refuses to select the optimal action with a probability of ε. This reduces the expressiveness of Q-Learning algorithm and increases the operation time. In order to solve the disadvantage of ε-greedy, the method of ε-decreasing[15] is proposed. In this method, the initial value of ε is large, and from the perspective of single iteration, the ε value increases with the increase of the number of steps in a single iteration, which enhances the ability of the agent to explore the new environment. From the perspective of the number of global iterations, the ε value decreases with the increase of the number of iterations to avoid over-exploration in the later stage. In addition, a new equilibrium idea ε-first[16] is proposed, which assigns the value of ε to 1 at the beginning of the main operation, and the agent is in a state of full exploration of the environment. After training for a certain number of times, ε is assigned to 0, and the agent is

Identify applicable funding agency here. If none, delete this text box.

in the state of fully utilizing the Q table. In view of this, I combine ε-decreasing with ε-first to optimize Q-Learning by using the characteristics of the attenuation oscillation curve.

We will propose a new optimization model. The balance between exploration and exploitation of Q-Learning algorithm is optimized based on oscillation attenuation curve(AOCQ-Learning). The time allocation of exploration stage and exploitation stage is controlled reasonably by using the oscillation attenuation characteristic of attenuation oscillation curve. In this paper, the change of ε value and the change of proportion time is mainly controlled by the attenuation oscillation curve and some constraint parameters. The algorithm is applied to the path optimization experiment, and it is found that it has better performance.

The rest of this article is organized as follows. In the second part, the theory of reinforcement learning is introduced. The third part introduces the AOCQ-Learning algorithm model in detail. The fourth part introduces the experimental part of the application of several algorithms in path planning. The fifth part is the conclusion of this paper.

## II. RELATED THEORY

### A. Reinforcement Learning

The core elements of reinforcement learning include: strategy, income, value function and environment. The agent constantly interacts with the environment, the environment gives feedback to the agent's actions in the form of reward signals, and through continuous iteration, the strategy is gradually formed. The agent accomplishes the goal according to the learned strategy. The specific structure is shown in Fig.1.

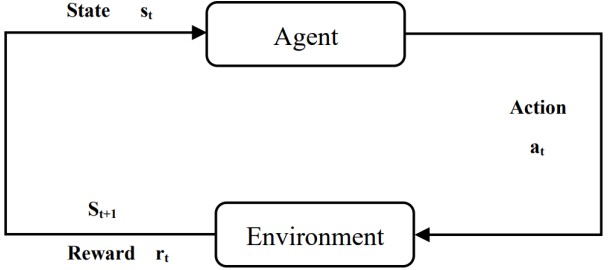

Fig. 1. Reinforcement learning structure

### B. Q-Learning Algorithm

Q-Learning algorithm is a value iterative algorithm, which approximates the optimal strategy by constantly updating the Q-table. It is one of the most widely used algorithms in reinforcement learning.

In the Q-Learning algorithm, the strategy established by the agent according to the income is represented by the Q(s , a) value table, which records the Q( s , a) value corresponding to all actions a under different states s, and guides the agent to complete the goal through the Q(s , a) value table. The update of

the Q(s , a) value follows the Bellman formula [1], as follows.

$$Q(s_t, a_t) \leftarrow Q(s_t, a_t) +$$
$$\alpha \left[ r_t + \gamma \max_a Q(s_{t+1}, a) - Q(s_t, a_t) \right] \quad (1)$$

In the formula, $Q(s_t, a_t)$ represents the Q value corresponding to the state-action ($s_t$, $a_t$); α represents the learning rate, indicating the degree to which the Q value learns from the updated Q value change; Gamma represents the discount rate, indicating how much the agent values future returns, and the discount rate $\gamma \rightarrow 0$ indicates that the agent is shortsighted and only cares about maximizing current return. The standard Q-Learning algorithm flow is as follows.

---

Algorithm1. Q-Learning

---

Step1. Initialize arbitrarily all Q(s,a) values;

Step2. Repeat (for each episode):

Step3. Define an initialization state $s_0$, s = $s_0$;

Step4. Repeat (for each step in the episode):

Step5. Select an action a ∈ A($s_t$) according to the policy;

Step6. Execute the action a, Observe the new state $s_{t+1}$, receive immediate reward $r_{t+1}$;

Step7. Update policy(Q-values table) according to:

$$Q(s_t, a_t) \leftarrow Q(s_t, a_t) + \alpha \left[ r_t + \gamma \max_a Q(s_{t+1}, a) - Q(s_t, a_t) \right]$$

Step8. $s_t = s_{t+1}$;

Step9. If $s_t$ is one of the goal states, episode += 1; Else go to Step5;

Step10. If the desired number of episodes have been investidated, END; Else go to Step3;

---

The original Q-Learning algorithm uses greedy strategy, that is, the action corresponding to the maximum Q value is selected, which is completely dependent on the strategy that has been learned. This situation will make the algorithm very easy to fall into the local optimal, and with the increase of learning time can not jump out. The ε-greedy strategy is introduced.

### C. ε -greedy-Q-Learning

In the Q-Learning algorithm based on ε-greedy strategy, a greed coefficient factor ε(0<ε≤1) is set. It means that when the agent chooses an action, there is an ε probability that it chooses randomly among all the actions, and there is a probability of 1-ε that it chooses the action with the greatest return. The algorithm indicates that the ε-probability of the agent does not depend on the learned strategy choice action, and if the agent falls into the local optimal, there is a certain probability of jumping out of the local optimal. However, Q-Learning algorithm based on ε-greedy strategy always explores with a

certain probability in the later stage, and the convergence time will increase. Therefore, an algorithm for dynamic attenuation of greed factor is proposed.

### D. ε -decreasing-Q-Learning

In the traditional Q-Learning algorithm based on ε-greedy strategy, the greed factor ε is a constant. Therefore, the ε-decreasing algorithm was adopted[15]. Where the greed factor ε changes with the number of iterations and the number of steps in a single iteration, the specific formula is as (2).

$$\varepsilon = \varepsilon_0 * 0.1^{\frac{episode}{step}} \tag{2}$$

In formula (2), the parameter "episode" represents the number of iterations, and "step" represents the number of steps in a single iteration. At the early stage of iteration, due to less understanding of the environment information, the greed factor $\varepsilon_0$ is set to a large size to ensure full exploration of the environment. From the perspective of single iteration, ε increases with the increase of the number of steps. With the increase of the number of iterations, the agent's mastery of environmental information more requires the agent to use the learned empirical knowledge for action selection, so the greed factor ε is in a gradual decline process.

### E. ε-first-Q-Learning

The idea of the ε-first method[16] is to initially set the value of ε to 1, so that the agent is in the full exploration state, and after a period of training, set the value of ε to 0, so that the agent is in the full exploitation environment state. The number of training acts is set according to the actual situation.

### III. AOCQ-LEARNING ALGORITHM

### A. Attenuation Oscillation Curve

In this paper, attenuation oscillation curve is introduced into Q-Learning algorithm, and its expression is shown as formula (3).

$$y = e^{W_1 T} \times \sin(W_2 \times T) \tag{3}$$

Balance the exploration phase and the exploitation phase by changing the thresholds [DOWN,UP]. Take formula (4) as follows.

$$\varepsilon = \begin{cases} 1, DOWN < y < UP \\ 0, other \end{cases} \tag{4}$$

When ε = 1, the algorithm is in the exploitation stage. When ε = 0, the algorithm is in the exploration phase. The balance between the exploration stage and the exploitation stage can be controlled by changing the value of the independent variable T and the change rate of the function value

y. The attenuation oscillation curve is shown in Fig. 2. When the agent passes through the area of the curve AOC indicated by the green bracketed arrow, the selection strategy of the action is random selection. Instead, the action selection strategy follows the Q table. As the number of iterations increases, the upper limit of switching threshold UP increases in the same direction as the number of iterations, which means that the random exploration time of the agent decreases. However, due to the lower limit of the switching threshold DOWN, the agent will also conduct some random exploration. If the algorithm does a lot of exploration in the early stage, that is, there are a lot of overflow times, the parameter DOWN will move down beyond AOC. In this case, the algorithm will converge completely in the later stage to avoid the selection of invalid actions.

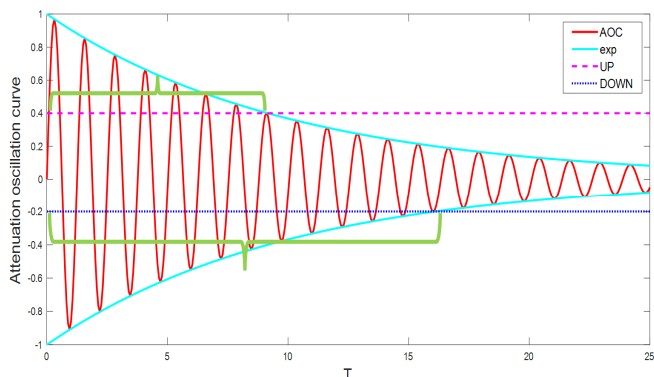

Fig. 2. Attenuation oscillation curve model

In the process of ε-decreasing, the ε value gradually decreases with the increase of the number of iterations, and the change of ε value in ε-first is first 1 and then 0, which is combined with the characteristics of the attenuation oscillation curve to form the AOCQ-Learning algorithm in this paper.

The attenuation oscillation curve is introduced into Q-Learning algorithm, and the algorithm is divided into exploration stage and exploitation stage by comparing function value and switching threshold. According to the iteration state of the agent, the time proportion of the exploration stage and the exploitation stage is adjusted. The original Q-Learning algorithm is improved as follows.

⑴ After initializing the Q(s,a) value table, in order to avoid blind random exploration by the agent, the greedy strategy (ε = 1) is used at the beginning stage. When the number of successful explorations is greater than the constant N, random exploration is added.

⑵ In each iteration of the algorithm, the early warning signal

K1 is set for the number of iteration steps. If the number of single steps exceeds K1, the exploration phase time will be reduced and the exploitation phase time will be increased.

（3）Step overflow limit signal K2(K1<K2) exists in each iteration of the algorithm. When the number of steps in a single iteration exceeds K2, the algorithm actively interrupts the exploration and starts the iteration again.

（4）Set parameters DOWN and UP to control the switch between the exploration phase and the exploitation phase. Specific AOCQ-Learning pseudocode such as Algorithm2:

---

**Algorithm2：AOCQ-Learning**

---

Step1: Initialize arbitrarily all Q(s,a) values; Initialize parameters;

Step2: Start with the initial state s0; Repeat (for each step in the episode):

Step3: If the number of reaching the goal K > N，go to Step4；Else，implement the traditional Q-Learning algorithm;

Step4: If DOWN < F(T) < UP：Select an action $a_o$ according to ε = 1，a=$a_o$；Else；Select an action $a_o$ according to ε = 0，a=$a_o$；T += t(const);

Step5: Execute the a，turn into next state s';

Step6: If the number of the step i >K1，Adjust the rate of the function variable 'T';

Step7: If the number of the step i > K2，i = K2，the lower threshold 'DOWN' moves downward，episode += 1，go to Step2;

Step8: If $s_{t+1}$ is goal，episode += 1，K += 1；Else，$s_{t+1}$ is obstacle or free field，get reward R，update Q table，go to Step3;

Step9: If episode < M(const)，go to Step2；Else termination loop;

---

TABLE. I. Parameter description

| Parameter | Function Description |
|---|---|
| UP | Switchover action Specifies the upper threshold for selecting a policy. It depends on the number of iterations. As the number of iterations increases, the corresponding upward movement. |
| DOWN | Lower limit of the switchover action selection policy. Related to step overflow in a single iteration. As the number of steps overflow of a single iteration increases, it corresponds to a downward movement. |
| K1 | Single iteration step number |
| | warning signal. If the number of steps in a single iteration exceeds the warning signal, change the rate of change of the independent variable 'T' of the attenuation oscillation curve. Shorten the time of random action selection and increase the time of action selection based on Q table. |
| K2 | Single iteration step overflow signal. If the number of steps in a single iteration exceeds the overflow signal, the iteration process is terminated actively. Lower limit threshold 'DOWN' corresponds to downward movement. |
| K | In the initial iteration, in order to avoid blind exploration in the early stage, the selection of actions is based on the Q table. When the number of iterations ['episode' > K] is reached, the AOC-QL algorithm is used. If the number of steps in a single iteration exceeds the overflow signal, the iteration process is terminated actively. Lower limit threshold 'DOWN' corresponds to downward movement. |

Where, the constant N represents the number of times that the Q table stage is fully exploited in the initial period; Parameters [DOWN, UP] represent the thresholds of the switch exploration stage and the exploitation stage in the attenuation oscillation curve. The upper threshold UP changes with the number of iterations episode, while the lower threshold DOWN is related to step overflow. The parameter T is the independent variable of the attenuation oscillation curve. By controlling the change rate of independent variable T, the proportion of exploration stage time and exploitation stage time in a single iteration can be controlled. Parameter K1 represents the early warning signal of the number of steps in a single iteration, which is related to the average number of steps in previous iterations; The parameter K2 represents the maximum limit on the number of steps in a single iteration; The episode parameter indicates the number of iterations. The parameters are described in TABLE. I.

*B. Parameter Selection*

In AOC-Q-Learning algorithm, by changing the growth of threshold values UP and DOWN and the change rate of time

variable T, the action selection strategy can be switched. Among them, the oscillation attenuation curve involves parameters $W_1$ and $W_2$.

The parameter $W_1$ controls the amplitude of the attenuation oscillation curve, as shown in Fig. 3. The smaller the value of W_1, the faster the amplitude of the attenuation oscillation curve decreases. Therefore, if the value of $|W_1|$ is smaller, the convergence time of the algorithm will be shorter, and the risk of the algorithm falling into local optimal will be increased.

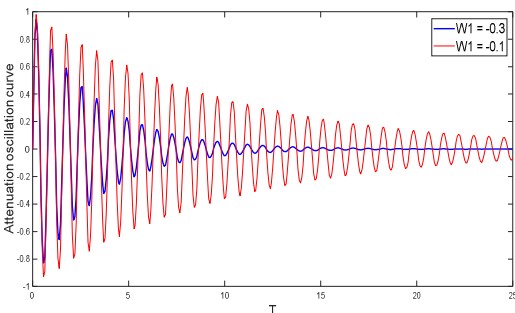

Fig. 3. Curve amplitude change effect

The parameter $W_2$ controls the oscillation frequency of the attenuation oscillation curve, as shown in Fig.4. The higher the value of $W_2$, the higher the oscillation frequency of the attenuation oscillation curve. Therefore, if the value of $W_2$ parameter is too small, the switching period between the exploration stage and the exploitation stage is too long, the overflow times of the algorithm in the early iteration stage will increase, and the algorithm will converge in advance and easily fall into the local optimal. However, if the $W_2$ parameter is too large, the switching period between the exploration stage and the exploitation stage is too short, which leads to slow convergence in the later iteration of the algorithm and increases the exploration time.

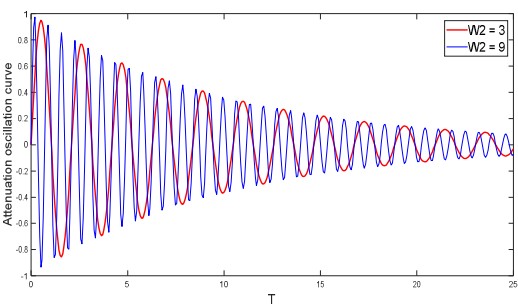

Fig. 4. Curve frequency change effect

## IV. EXPERIMENT

### A. Experimental Design

Based on the different strategies of how to balance the exploration stage and exploitation stage of Q-Learning algorithm, two 20*20 grid maps, namely environment 1 and environment 2, are established to represent the environment model in the algorithm, and each grid represents the state of the agent, as shown in Fig.5. Among them, the black square represents the obstacle, the yellow square represents the starting point, and the purple square represents the end point, and this environmental information is unknown to the agent. The action space of the agent is: up, down, left, and right. The grid coordinates correspond to the state of the agent.

When the agent encounters an obstacle, the agent returns to the upper state, and the environment returns the reward value r = -1; When the agent encounters the target, the agent completes an iteration, and the environment returns the reward value r = 1; If the agent does not encounter an obstacle or target, the environment returns the reward value r = 0. In this experiment, the number of iterations episode = 200, the learning rate α = 0.01, and the discount factor γ = 0.9. For the attenuation oscillation curve, $W_1 = -1$, $W_2 = 7$.

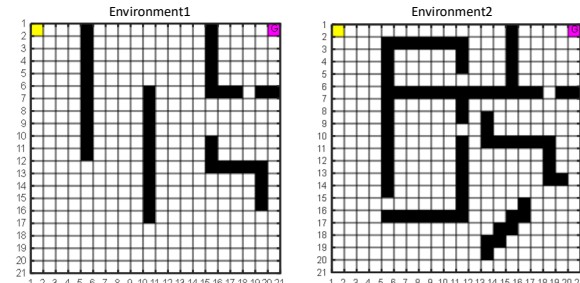

Fig. 5. Environment

### B. Comparative Analysis

In the experiment, the Pure-Q-Learning algorithm, ε-greedy-Q-Learning algorithm and AOCQ-Learning algorithm are tested in environment 1 and environment 2 respectively, and 10 groups of experimental data are taken from each of the four algorithms. Compare their average deviations.

(1) As shown in Fig.6 and Fig.9, the iterative process of the average value of 10 groups of experimental data of the three algorithms is displayed. In the whole iteration process of AOCQ-Learning, the iterative effect of the early path optimization is better than the other two algorithms, and the convergence of the later iteration is late, but the path length sought is better than the other two algorithms.

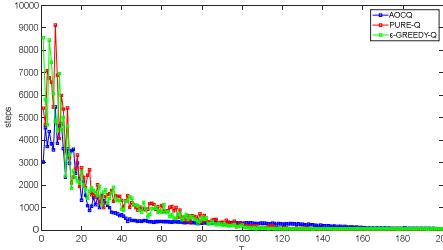

Fig. 6. Algorithm iteration process in Environment 1

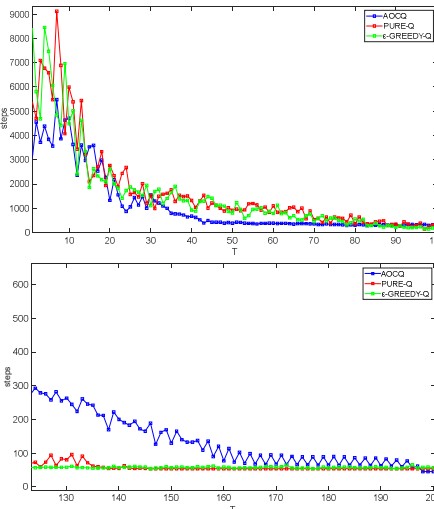

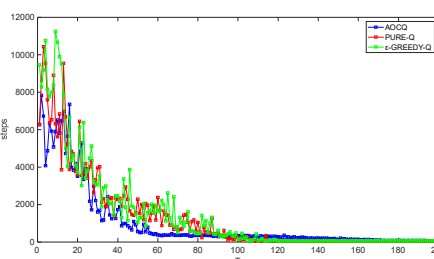

Fig. 7.Partial enlargement of Fig.6

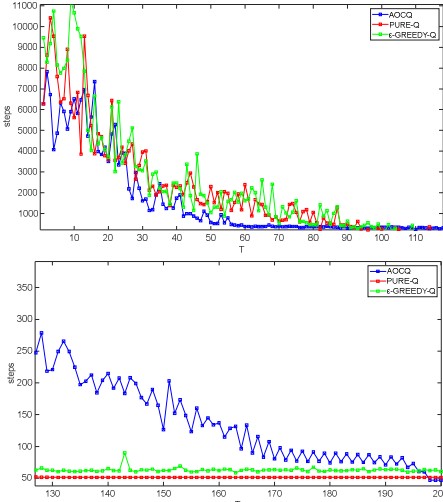

Fig. 8. Algorithm iteration process in Environment 2

（2）As shown in TABLE.II, the Average learning time from the start state to the target state is called the Average learning time (Alt); The Average optimal steps (Aos) of the agent from the start state to the target state; The Average exploration steps from the initial state to the target state are called Average exploration steps (Aes). The number of times the agent successfully explores to the optimal path is represented by Success(%). Standard deviation (Std) represents the standard deviation between the optimal path explored by different algorithms and the actual optimal path. By comparing the learning results of the optimal path sought by the Pure-Q-Learning algorithm, ε-greedy-Q-Learning algorithm and AOCQ-Learning algorithm in environment 1 and environment 2, it can be seen that the AOCQ-Learning algorithm has a faster convergence time. Shorter paths can be found. Moreover, the standard deviation

between the path found by AOCQ-Learning and the optimal path is smaller, which indicates that AOCQ-Learning has better exploration ability.

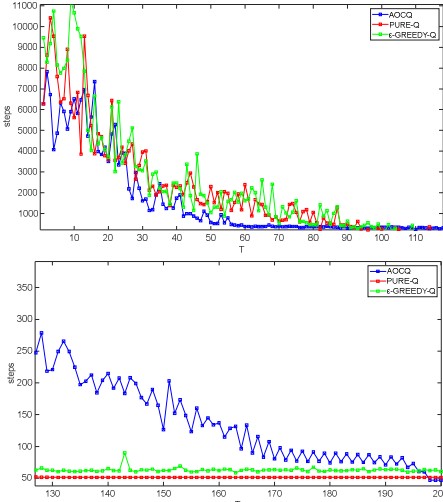

Fig. 9. Partial enlargement of Fig.8

TABLE. II. Comparison of experimental results

| Methods | Alt | Aos | Aes | Success(%) | Std |
|---|---|---|---|---|---|
| *Environment1* | | | | | |
| Pure-Q-L | 334.23s | 53.6 | 178055.7 | 10 | 5.09 |
| ε-greedy-Q-L | 299.45s | 53.9 | 166443.2 | 20 | 5.32 |
| AOCQ-L | 210.36s | 45.6 | 131462.8 | 90 | 1.89 |
| *Environment2* | | | | | |
| Pure-Q-L | 466.44s | 51.2 | 264891.4 | 0 | 9.47 |
| ε-greedy-Q-L | 500.27s | 58.4 | 297229.0 | 20 | 7.93 |
| AOCQ-L | 318.41s | 48.0 | 205962.7 | 90 | 1.89 |

In summary, the Pure-Q-Learning algorithm, ε-greedy-Q-Learning algorithm and AOCQ-Learning algorithm are applied to path optimization. The AOCQ-Learning algorithm can seek a better path with shorter time and fewer steps.

The characteristics of the attenuation oscillation curve are well qualified for the constraint parameters of Q-Learning algorithm mentioned in this paper. With the increase of information obtained by the agent through interaction with the environment, due to the attenuation characteristic of the curve, the agent gradually increases the time of exploration and speeds up the convergence rate. However, due to the oscillatory nature of the curve, while the agent is gradually converging in the exploitation, the accompanying mutation of the algorithm leads to the probability that the agent is in the exploration, which prevents the algorithm from falling into the local optimal with a certain probability.

When the attenuation oscillation curve is introduced into Q-Learning algorithm, the amplitude and frequency of the attenuation oscillation curve need to be considered. Through experimental observation, the amplitude is related to the convergence speed of the algorithm, and the oscillation frequency is related to the iteration time of the algorithm. When the environment is relatively simple, the amplitude parameter is reduced, the oscillation frequency parameter is reduced, the algorithm exploitation mode occupies a large proportion, and the operation time is saved. When the environment is relatively complex, the amplitude parameter and oscillation frequency parameter are increased, the algorithm exploration mode occupies a larger proportion, and the environmental information obtained is increased, which avoids falling into the local optimal problem with a certain probability. Among them, the parameter adjustment amplitude is related to the change rate of the independent variable T of the oscillation attenuation function. Therefore, in the AOCQ-Learning algorithm mentioned in this paper, simulation experiments should be carried out according to the actual environment to determine the stable interval of mutual equilibrium of each parameter.

## V. CONCLUSION

Through the introduction of oscillation attenuation curve, the action selection strategy of traditional Q-Learning algorithm is improved, and AOCQ-Learning algorithm is proposed. By switching between the exploration phase and the exploitation phase, it can help the agent to find a better path in a shorter time with fewer iteration steps. In addition, AOCQ-Learning algorithm can continue to be optimized through more reasonable parameter Settings.

## ACKNOWLEDGMENT

This work was supported in part by the National Natural Science Foundation of China (grant nos. 52131101 and 51939001), the Liao Ning Revitalization Talents Program (grant no. XLYC1807046), and the Science and Technology Fund for Distinguished Young Scholars of Dalian (grant no. 2021RJ08).

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
