# OpenReview forum: "An improvement of Q-Learning based on attenuation oscillation curve for path planning"
_IEEE.org/ICIST/2024/Conference — IEEE ICIST 2024 Conference Submission_

### Official Review · Reviewer_ZU8m · 2024-08-21
**This paper proposes a model based on Q-Learning algorithm to balance the exploration stage and the exploitation stage. Meanwhile, the experiment verified the effectiveness of the method. However, the following comments should be considered in the revision.**

**Rating:** 7
**Confidence:** 3

**Review:**

Question 1:
How does the attenuation oscillation curve (AOC) specifically improve the balance between exploration and exploitation in the Q-Learning algorithm compared to classical methods, and what theoretical foundations support its effectiveness?
Question 2:
What are the computational complexities introduced by incorporating the attenuation oscillation curve into the Q-Learning algorithm, and how do these complexities affect the overall performance and scalability of the proposed approach?
Question 3:
Please provide a more detailed analysis of the parameter tuning for the attenuation oscillation curve? How sensitive is the algorithm's performance to variations in these parameters, and what guidelines are provided for setting them in different types of environments?

---

### Official Review · Reviewer_vJdN · 2024-08-22
**The paper is logically clear, the simulation results are credible. The following suggestions need to be considered:**

**Rating:** 6
**Confidence:** 3

**Review:**

1. What impact will the inability to achieve this balance between exploration and exploitation of the Q-Learning algorithm have, and how does the proposed new optimization model achieve this balance?
2. How does the introduction of oscillation attenuation curves help with the execution of the Q-learning algorithm?
3. In the Abstract, the logic of the sentence “based on Q-Learning algorithm, attenuation oscillation curve is introduced into it” seems unreasonable.
4. Some grammar issues need to be checked and modified, such as "how to exploration and exploitation", etc.

---

### Decision · Program_Chairs · 2024-09-06

Accept (Oral)